# Reactive Oxygen Species Mediate Transcriptional Responses to Dopamine and Cocaine in Human Cerebral Organoids

**DOI:** 10.3390/ijms242216474

**Published:** 2023-11-18

**Authors:** Thomas T. Rudibaugh, Samantha R. Stuppy, Albert J. Keung

**Affiliations:** Department of Chemical and Biomolecular Engineering, North Carolina State University, Raleigh, NC 27606, USA; ttrudiba@ncsu.edu (T.T.R.); srstuppy@ncsu.edu (S.R.S.)

**Keywords:** cerebral organoids, dopamine, cocaine, reactive oxygen species (ROS), RNA-seq, acute-chronic, epigenetics

## Abstract

Dopamine signaling in the adult ventral forebrain regulates behavior, stress response, and memory formation and in neurodevelopment regulates neural differentiation and cell migration. Excessive dopamine levels, including those due to cocaine use in utero and in adults, could lead to long-term adverse consequences. The mechanisms underlying both homeostatic and pathological changes remain unclear, in part due to the diverse cellular responses elicited by dopamine and the reliance on animal models that exhibit species-specific differences in dopamine signaling. In this study, we use the human-derived ventral forebrain organoid model of Xiang–Tanaka and characterize their response to cocaine or dopamine. We explore dosing regimens of dopamine or cocaine to simulate acute or chronic exposure. We then use calcium imaging, cAMP imaging, and bulk RNA-sequencing to measure responses to cocaine or dopamine exposure. We observe an upregulation of inflammatory pathways in addition to indicators of oxidative stress following exposure. Using inhibitors of reactive oxygen species (ROS), we then show ROS to be necessary for multiple transcriptional responses of cocaine exposure. These results highlight novel response pathways and validate the potential of cerebral organoids as in vitro human models for studying complex biological processes in the brain.

## 1. Introduction

Dopamine signaling in the ventral forebrain is crucial in regulating behavior, stress response, and memory formation in adults [1,2,3,4]. Furthermore, it plays a significant role in neural differentiation and cell migration during neurodevelopment [5,6]. Excess levels of extracellular dopamine can lead to adverse long-term consequences, such as addiction formation, memory loss, premature neural differentiation, and neuroinflammation [5,7,8,9,10,11,12,13,14]. Despite its importance, the mechanisms underlying these alterations remain unclear, and, in particular, our understanding of dopamine signaling in human systems is even further limited [12,14,15,16,17,18,19].

Previous studies have shown human pluripotent stem cells can be differentiated in 2-D towards ventral forebrain neurons expressing functional dopamine receptors [20,21,22,23,24,25], and these neurons can even recapitulate complex phenomena such as a gene desensitization following chronic dopamine exposure [26]. Recently, 3-D cerebral organoids have been developed to model neurodevelopment, cell migration patterns of different brain regions [27,28], and stem cell and neuronal responses to external stimuli, including substances of abuse [29,30,31,32]. It will be important to characterize organoid responses more deeply to stimuli, to help identify underlying pathways involved in dopamine function, to determine the phenotypes that organoids can replicate well, and identify those areas for further improvement.

In this work, we leverage the previously described Xiang–Tanaka ventral forebrain organoids to model ventral forebrain neurodevelopment and investigate their responses to acute and chronic dopamine exposure using a combination of next-generation sequencing along with Ca^2+^ and cAMP measurements [28]. Through our analysis, we observe immune-related transcriptional responses and identify novel response pathways activated following dopamine exposure. Using small molecule inhibitors, responses are found to act at least in part through reactive oxygen species and in part through dopamine receptor signaling. Finally, we observe some similarities in the responses to chronic cocaine exposure. These findings demonstrate the utility of ventral forebrain organoids as an in vitro human platform for studying complex biological processes and their potential mechanisms.

## 2. Results

### 2.1. Ventral Forebrain Organoids Exhibit Ca^2+^ and Intracellular cAMP Responses to Dopamine

Xiang–Tanaka and colleagues described the generation of human medial ganglionic eminence organoids (hMGEOs) through the activation of the sonic hedgehog (SHH) pathway to achieve ventral forebrain identity [28]. In this work, we characterize the functional and transcriptomic responses of these organoids to dopamine and cocaine. We first confirm the generation of ventral forebrain identity. By day 30 (D30), *GSX2*+/*SOX2*+ progenitor cells arise, suggesting an early ventral forebrain fate [21,33] (Figure 1A). By D90, we observe robust expression of *GABA*+/*MAP2*+/*CTIP2*+, indicating development of GABAergic neurons (Figure 1B), and of medium spiny neuron (MSN) marker *DARPP32* (Figure 1C). In addition, these organoids show higher expression of multiple MSN-related genes compared to stem cells, further suggesting a ventral forebrain identity (Appendix A). In-depth compositional analysis of these hMGEOs, demonstrating a ventral forebrain identity and enrichment for GABAergic neurons, have both been performed by Xiang–Tanaka and colleagues [28,34]. After confirming a ventral forebrain fate, we assess the presence of calcium activity, observing spontaneous transients through Fluo-4 Ca^2+^ imaging in D90 organoids (Appendix A). These transients are inhibited by the addition of the sodium channel blocker tetrodotoxin (TTX), indicating the Ca^2+^ transients are likely to be dependent on neuronal activity. Taken together, these results show the differentiation of stem cells into relatively mature and functionally active ventral forebrain organoids [28].

Dopamine is a major neurotransmitter impinging on ventral forebrain neurons. In previous work, 2-D ventral forebrain neurons responded electrophysiologically and transcriptionally to dopamine [5,6,24,26]. In addition, there is evidence to suggest cultured neurons can exhibit neuroplastic alterations following chronic dopamine exposure [26]. We ask if this organoid model similarly exhibits a response to dopamine and if this response changes following chronic dopamine exposure. It is unclear, however, what dopamine concentration correlates to in vivo concentrations at the synaptic cleft. Previous work in mice suggests dopamine concentrations anywhere from 100 nM–100 μM, while human cell culture methods have used higher concentrations of up to 10 mM dopamine to elicit a response [26,35,36,37]. Therefore, we treat D90 organoids to either an acute or chronic dose of low concentration (1 μM) or high concentration (1 mM) dopamine along with a vehicle control (Figure 1D,E and Appendix A, Appendix A). Ideally, these organoids would respond to the more physiologically relevant dose (1 μM); however, it is possible that the higher concentration is needed to observe a response. Surprisingly, neither dopamine concentration nor chronic exposure causes a change in Ca^2+^ transient frequency or amplitude when measured at the population level (Appendix A).

One possible explanation is that dopamine causes opposing responses in different cell types that average out at the population level. D1-type dopamine receptor stimulation usually causes an increase in action potential frequency, whereas D2 receptor stimulation usually leads to a decrease, although there are exceptions [38,39,40]. While it is technically challenging to track Ca^2+^ transients simultaneously with identifying the neuronal subtype of each cell, we can longitudinally compare the Ca^2+^ transient frequency and amplitude of each individual cell before and after dopamine- or vehicle control exposure. We limit our analysis to active cells that exhibit at least one measurable calcium transient during the 10-min baseline measurement. The results show over 50% of cells treated with vehicle controls have the same Ca^2+^ transient frequency and amplitude as their baseline, while only 20–30% of dopamine-treated cells maintain the same frequency and amplitude (Figure 1E). This suggests that, of the cells measured, roughly 25% are exhibiting a change in Ca^2+^ transients that can be directly attributed to dopamine. Both acute 1 µM and 1 mM dosing regimens lead to a similar phenotype, where many individual cells exhibit a decrease in Ca^2+^ transient frequency and an increase in amplitude following dopamine exposure. In addition, we observe a robust response following 1 μM dopamine exposure, suggesting this lower concentration is enough to drive a cellular response in organoids. Thus, for all future dosing experiments, we used 1 μM dopamine.

In addition to tracking Ca^2+^ transients, we can also assess how intracellular cAMP changes in response to dopamine. Dopamine signaling is known to drive either an increase or decrease in intracellular cAMP depending on the neural subtype [12,41,42,43], and cAMP leads to a transcriptional response in ventral GABAergic neurons, ultimately contributing to many of the downstream neuroplastic changes associated with chronic drug exposure [3,10,44,45]. To our knowledge, cAMP and complex downstream responses have not been previously described in stem cell-derived systems. Thus, we expose organoids to acute and chronic 1 µM dopamine and measure the levels of intracellular cAMP (Figure 1D,F and Appendix A, Appendix A). Reassuringly, dopamine causes an increase in the amount of intracellular cAMP, and this response is dampened in the chronically dosed samples (Figure 1F and Appendix A), indicating a potential desensitization phenotype. 

### 2.2. Transcriptional Analysis Reveals Immune-Related Response to Acute and Chronic Dopamine

While Ca^2+^ and cAMP measurements indicate functional responses to dopamine, transcriptomic changes could provide deeper insight into how long-term neuroplastic alterations might arise [8,14,46,47]. In addition, transcriptomic changes caused by chronic dopamine overexposure are implicated in addiction formation in adults and neurodevelopmental alterations in utero [5,8,48,49]. Therefore, we expose D90 organoids to acute and chronic doses of 1 μM dopamine followed by bulk RNA sequencing and differential gene expression (DGE) analysis relative to D90 vehicle control (Figure 2 and Appendix A, Appendix A). To simulate an acute dose, we expose our organoids for 1 h. As the neurons are localized to the outer periphery, and the interior of the organoids have relatively few cells due to nutrient limitations (Appendix A), we expect the majority of neurons and cells in general to be exposed to dopamine. In addition, we compare our organoids to 12pcw striatum neurons as previous analysis suggests these organoids contain MSNs [34] (Appendix A), and match that fetal timepoint best across multiple types of organoids [50,51]. As expected, transcriptomic analysis shows a similar expression of ventral forebrain markers suggesting these organoids are a good model for the developing ventral forebrain (Appendix A, Appendix A). A notable exception is an increased expression of more ventral forebrain markers in our organoids along with higher abundance of GABAergic receptors. As these organoids were initially developed to recapitulate the more ventral forebrain region, higher expression of *DLX1* and *NKX2* is expected and suggests these organoids also contain more ventral neurons. Collectively, these compositional results further support the use of these organoids to model the ventral forebrain.

Following composition analysis, we perform differential gene expression between the acute and chronically dosed samples and the vehicle controls. Chronic dopamine exposure causes an increase in the number of differentially expressed genes (DEGs) compared to the acutely dosed samples (Figure 2B). Among the top 20 downregulated genes in the acutely dosed samples are many genes in the HOX protein family, which play a role in cell migration and apoptosis during neurodevelopment [52,53] (Figure 2C, left). Among the top upregulated DEGs in the acutely dosed organoids are the anti-inflammatory cytokines, *IL1R1* and *TGFβI,* along with two genes linked with a pro-inflammatory immune response, *FCN1* and *SPON2.* In addition, we observe activation of the ECM-related genes *SERPINE1*, and *FN1* [54,55,56,57,58,59,60]. Among the top 20 upregulated genes for the chronically dosed organoids are pro-inflammatory immune response markers, *S100A9* and *FGR*, along with anti-inflammatory immune response markers, *HSD11B1* and *ITGB2* [61] (Figure 2C, right). In addition, we observe upregulation of multiple collagen proteins.

Comparison of the DEGs from the acute and chronic conditions shows 21 genes shared between both conditions, and many of these genes are again linked with the ECM and immune response including *IL1R1*, *TGFβI*, *SERPINE1*, and *FN1* among others (Figure 2D, Appendix A). Gene set enrichment analysis (GSEA) also suggests both the acute and chronic exposures are acting through similar upstream regulators related to either a pro-inflammatory immune response including lipopolysaccharide, *IL6*, *IL1B*, and *TNF* or through an anti-inflammatory immune response including *TGF-β1*, *EGF*, and *STAT3* [62,63] (Figure 2E).

### 2.3. Organoid Response to Dopamine Driven by Reactive Oxygen Species 

Alterations in the expression of immune-related genes and HOX genes suggested dopamine may not only be acting through canonical neuronal signaling pathways. In particular, dopamine has been shown to activate cellular immune responses by both binding to dopamine receptors directly and through dopamine metabolism by monoamine oxidases [12,13], and there is ample evidence that non-immune cells such as neurons activate immune responses [64]. As another example, dopamine metabolism by microglia leads to an increase in both intra- and extra-cellular reactive oxygen species (ROS), which in turn leads to oxidative stress and activation of immune cells [65,66]. This immune response induces ECM remodeling, including the increased expression of multiple collagen proteins especially collagen I proteins [67]. Despite lacking microglia, we observe similar responses in our samples. Therefore, we hypothesize that ROS may be mediating the transcriptional responses we are observing to dopamine.

To test this hypothesis, we measure the endogenous ROS levels of D90 organoids exposed to acute and chronic dopamine. Both samples show higher levels of endogenous ROS (Figure 3A). We also measure the levels of cAMP following dopamine exposure in the presence of either the dopamine receptor antagonists SCH-23390 and sulpiride or the ROS inhibitor acetylcysteine (Figure 3B and Appendix A). We observe dopamine receptor antagonists show some ability to blunt the increase in intracellular cAMP while the ROS inhibitor dramatically reduces the intracellular cAMP response. Similarly, we observe that acetylcysteine also substantially blunted the transcriptomic response of 1 μM dopamine down to control levels in D90 organoids (Figure 3C–F and Appendix A, Appendix A). To ensure acetylcysteine alone is not driving this phenotype, we included controls both with and without acetylcysteine exposure (Figure 3D).

GSEA of the DEGs from the chronically dosed samples once again reveals that, among the top upstream regulators are multiple pro- and anti-inflammatory cytokines including *IL5*, *IL6*, *EIFAK3*, and *TGFβI* among others [68,69] (Figure 3E). In addition, among the top five regulated pathways are the unfolded protein-response pathway and endoplasmic reticulum stress-response pathways (Figure 3F). One key signature of oxidative stress is an increase in unfolded proteins accumulating in the ER [70,71]. Collectively, these results suggest a potential influence of dopamine on neurodevelopment through ROS.

### 2.4. ROS Also Mediates the Transcriptomic Response to Cocaine

Cocaine is related to dopamine signaling in that it causes an increase of dopamine in the synaptic cleft through inhibiting the presynaptic dopamine transporter [8,48]. Interestingly, prenatal cocaine exposure has also been linked to oxidative stress in the cerebral cortex, leading to premature neural differentiation [29,32,70,72]. There is evidence to suggest cocaine acts through a similar mechanism in the developing ventral forebrain as well [73]. Therefore, we expose D90 ventral forebrain organoids to chronic doses of 1 μM cocaine with and without acetylcysteine (Figure 4 and Appendix A, Appendix A). The presence of acetylcysteine dramatically reduces the number of DEGs (Figure 4B). Among the top upregulated genes following cocaine exposure, we observe upregulation of multiple HOX family genes (Figure 4C). In addition, we also observe upregulation of multiple immune-related genes including *IL1R1*, *SERPINA3*, and *CYP1B1* (Figure 4D) [74]. GSEA suggests many of the upstream regulators identified following dopamine exposure are also activated by cocaine, and this activation is ROS-dependent (Figure 4E).

Given similarities in ROS mediating transcriptomic responses to dopamine and cocaine, we ask how similar the organoids respond transcriptomically to these two compounds. A comparison of DEGs reveals almost 50% of the DEGs in the dopamine-dosed samples are present in the cocaine-dosed samples, and many of them are immune-related genes (Figure 4F, Appendix A). Only one of these genes, *HSPA6*, appears as DEG in the cocaine samples with ROS inhibited, suggesting the activation of these immune transcripts is ROS-dependent for both the dopamine- and cocaine-dosed samples. GSEA indicates many of the top regulated pathways are the same for both the cocaine- and dopamine-dosed samples, and ROS inhibition dampens these pathways (Figure 4G). Furthermore, organoids demonstrate downregulation of netrin signaling, responsible for ECM organization and cell migration during neurodevelopment, in both our dopamine- and cocaine-dosed samples. We also observe upregulation of the immunogenic cell signaling pathway in the cocaine- and dopamine-dosed samples, which is associated with stress-induced apoptosis (Figure 4G). These pathways are both reduced in the acetylcysteine conditions, further suggesting the immune-related and ECM responses caused by dopamine and cocaine are ROS-mediated. A similar trend is observed in the upstream regulators (Appendix A).

## 3. Discussion

While it has been widely accepted that dopamine’s effects are primarily driven by dopamine receptor binding and subsequent intracellular signaling in adult neurons, recent evidence from mouse studies suggests that dopamine-mediated reactive oxygen species (ROS) also contribute significantly to neurodevelopment [65,75,76], and that dopamine may regulate cell cycles and differentiation of stem and progenitor cells [5]. Our results using human ventral forebrain organoids suggest that dopamine-mediated ROS may hold a similar role in human neurodevelopment, although these findings would need to be validated in vivo.

Human stem cell-derived models are known to be heterogeneous, and therefore a limitation of this study is the lack of cell-type specific resolution. However, this study provides results that suggest several interesting avenues for investigation. First, the transcriptomic responses involving immune, inflammation, ECM, and HOX genes suggest that dopamine and cocaine likely act on multiple different cell types and therefore future work should begin to dissect these different effects in both cell culture and animal models. Our own single-cell sequencing of these organoids suggests a diversity of cell types, although GABAergic neurons are the plurality [34]. Related, the diffusion of dopamine and cocaine into these 3-D organoids may affect which cell types are exposed to the compounds. As a well-known phenomenon with ventral forebrain organoids, we have observed the majority of cells are located along the periphery of the organoids due to the relative lack of nutrients reaching the organoid core. Furthermore, neurons typically reside on the periphery relative to progenitor cells. Therefore, it is possible the responses observed here are dominated by neurons, but future work—with improved spatiotemporal control over compound delivery and with single-cell resolution containing sufficient read depth to perform differential gene expression analysis—would be useful.

Second, our results indicate this complexity extends to multiple mechanisms of action, where dopamine can act through dopamine receptors but also through other as-yet unknown pathways that may include metabolism or oxidative stress induced by byproducts of dopamine and cocaine.

Third, related to the heterogeneity of human cerebral organoids, these organoids typically lack immune cells of the brain, such as microglia. Thus, the upregulation of inflammatory pathways in response to cocaine and dopamine exposure may reflect how non-immune cells can sense and respond to inflammatory conditions to help prime the immune system. This observation underscores the possibility that cells within the brain may assume supportive roles or mutually influence the immune system. These organoids could potentially serve as valuable models for studying inflammation, especially when combined with emerging organoid models that incorporate microglia [77]. Such combinations offer a promising avenue for unraveling the intricate interactions between non-immune and immune cells in the developing brain. Follow-up studies employing comprehensive inflammatory and hyperinflammatory assays will be crucial to deepen our understanding of the specific immune processes at play within these organoids.

Fourth, there are specific pathways and phenotypes that might be explored based upon our results. For example, among the top regulated pathways responsive to dopamine was the downregulation of multiple synapse-related pathways including synaptogenesis signaling and synaptic long-term potentiation. Additionally, we observe downregulation of CREB signaling and calcium signaling. Notably, previous research has associated prenatal cocaine exposure with reduced synapse formation [70,78,79]. As another example, premature cocaine exposure has been linked with decreased cell migration in utero and this response has been associated with downregulation of *BDNF* [57,78,80]. Interestingly, our GSEA identified *BDNF* as downregulated in cocaine-dosed organoids (Appendix A) [78,81]. As a final example, there is an intriguing link between the ECM and immune/inflammation-related genes that we observe altered by dopamine exposure. One possible explanation for increased collagen expression is activation of anti-inflammatory macrophages, which when activated leads to ECM deposition [82,83,84]. Given the ability of organoids to generate multiple cell types, this model offers an opportunity to understand these potential cell-type specific responses as well as their collective interactions.

## 4. Materials and Methods

### 4.1. hESC Cell Lines

H9 and H1 hESCs (WA09 and WA01; WiCell, Madison, WI, USA) were grown in E8 media (Thermo Fisher Scientific, Waltham, MA, USA) in 96-well culture dishes (Greiner Bio-One, Monroe, NC, USA) coated with 0.5 µg/mL Vitronectin (Thermo Fisher Scientific). Cells were passaged every 3–5 days as necessary using 0.5 mM EDTA (Thermo Fisher Scientific). All staining and qRT-PCR experiments included in Figure 1 and Appendix A were carried out on H1 and H9 stem cells. Due to cost limitations, all sequencing experiments were limited to H9 stem cells.

### 4.2. Organoid Culture

The Xiang ventral forebrain protocol organoids were generated as previously described with small modifications [28]. Stem cells were grown to 75% confluency before dissociation into a single-cell suspension using EDTA and Accutase (Fisher Scientific, Hampton, NH, USA). We seeded 9000 cells in a 96-well plate with low attachment U bottom (VWR, Radnor, PA, USA) in an induction medium supplemented with 50 µM Y-27632 (LC Labs, Woburn, MA, USA) and 5% *v*/*v* heat-inactivated fetal bovine serum (Corning, Corning, NY, USA). Induction media contained DMEM-F12 (Gibco, Waltham, MA, USA), 15% *v*/*v* knockout serum replacement (Thermo Fisher Scientific), 1% *v*/*v* MEM-NEAA (VWR), 1% *v*/*v* Glutamax (Gibco), 7 µL/L β-mercaptoethanol (Amresco, Solon, OH, USA), 100 nM LDN193189 (Sigma-Aldrich, St. Louis, MO, USA), 10 μM SB431452 (Abcam, Cambridge, UK), and 2 μM XAV939 (Sigma-Aldrich). After 48 h, half of the media was replaced with induction media containing Y-27632 and without heat-inactivated FBS. After another 48 h, half the media was again removed and replaced with induction media without Y-27632. On days 6–10, organoids were cultured in a 37 °C tissue culture incubator with 5% CO_2_ and regular half-media changes were performed every 48 h.

After day 10, organoids were transferred to a deep-bottom tissue culture-treated 10 cm plate (Thermo Fisher) containing differentiation media supplemented with 100 ng/mL SHH (VWR) and 1 µM purmorphamine (VWR). Differentiation media contained DMEM-F12, 0.15% *w*/*v* Dextrose (Sigma-Aldrich), 7 µL/L β-mercaptoethanol, 1% *v*/*v* N2 supplement (Thermo Fisher), 1% *v*/*v* Glutamax, 0.5% *v*/*v* MEM-NEAA, 0.025% insulin solution (VWR), and 2% *v*/*v* B27 supplement without vitamin A (Thermo Fisher). Cell culture plates were placed on a shaker in the tissue culture incubator and rotated at 70 rpm/min. On day 14 the media was removed and replaced with fresh differentiation media supplemented with SHH and purmorphamine. On day 18, the differentiation media was removed and replaced with maturation media containing vitamin A. Maturation media consisted of a 1:1 mixture of DMEM-F12 and neurobasal media (VWR), along with 0.5% *v*/*v* N2 supplement, 1% *v*/*v* B27 supplement with vitamin A (Thermo Fisher), 1% *v*/*v* Glutamax, 0.5% *v*/*v* MEM-NEAA, 0.025% *v*/*v* human insulin solution, 3.5 µL/L β-mercaptoethanol, and 1% *v*/*v* penicillin/streptomycin (VWR). In addition, 20 ng/mL of BDNF (Peprotech, Cranbury, NJ, USA) and 20 ng/mL GDNF (Peprotech) were added to the media. From day 18, organoids were cultured on the orbital shaker with weekly media changes in maturation media.

### 4.3. Cryosectioning and Immunohistochemistry

Tissues were fixed in 4% paraformaldehyde (Sigma-Aldrich) for 15 min at 4 °C followed by three 10-min PBS washes (Gibco). Tissues were placed in 30% sucrose overnight at 4 °C and then embedded in 10% gelatin/7.5% sucrose (Sigma-Aldrich). Embedded tissues were flash-frozen in an isopentane (Sigma-Aldrich) bath between −50 and −30 °C and stored at −80 °C. Frozen blocks were sectioned (Thermo Fisher) to 30 µm. For immunohistochemistry, sections were blocked and permeabilized in 0.3% Triton X-100 (VWR) and 5% normal donkey serum (VWR) in PBS. Sections were incubated with primary antibodies in 0.3% Triton X-100, 5% normal donkey serum in PBS overnight at 4 °C (Appendix A). Sections were then incubated with secondary antibodies in 0.3% Triton X-100, 5% normal donkey serum in PBS for 2 h at RT, and nuclei were stained with 300 nM DAPI (Invitrogen, Waltham, MA, USA) (Appendix A). Slides were mounted using ProLong Antifade Diamond (Thermo Fisher). Images were taken using a Nikon AR confocal laser scanning microscope (Nikon).

### 4.4. Real Time Quantitative PCR

D90 Organoids were washed three times in ice-cold PBS. Total RNA was isolated using Direct-zol RNA MicroPrep Kit (Zymo Research, Irvine, CA, USA) according to the manufacturer’s protocol. RNA samples were collected in 2 mL RNAse-free tubes and chilled on ice throughout the procedure. cDNA synthesis was performed using 1 µg of total RNA and the iScript Reverse Transcription Kit (BIO-RAD, Hercules, CA, USA) according to the manufacturer’s protocol. Real-time PCR was performed using the SYBR Green Supermix (BIO-RAD) according to the manufacturer’s protocol. Gene expression was compared to the reference gene GAPDH and ΔΔCq values were found by comparing against undifferentiated stem cell controls (Appendix A). For each qRT-PCR condition, five independent biological replicates of 2–3 organoids per replicate were collected.

### 4.5. Live Cell Imaging

Live imaging was performed using the Nikon AR confocal laser scanning microscope equipped with temperature and CO_2_ control. For calcium imaging, Fluo-4 direct (Life Technologies, Carlsbad, CA, USA) was prepared according to the manufacturer’s protocol. Whole organoids, due to their three-dimensional nature, can exhibit floating movements during imaging sessions. Therefore, the organoids were dissociated and replated in 2-D to reduce variability in fluorescence intensities due to inadvertent spatial motions and enhance the consistency of imaging results. D83 organoids were dissociated using Accutase and plated on reduced growth factor Matrigel (Corning) coated 24-well plates. Cells were cultured for 1 week in maturation media with 48-h media changes before dosing experiments were conducted. Cells were incubated with Fluo-4 60 min prior to the start of imaging. Frames were taken every 8 s for 10 min using a 10X objective. Following baseline measurement, plated cells were incubated in dopamine, either 1 µM or 1 mM, 1 µM of D1 agonist SKF-81297 (Tocris Biosciences, Bristol, UK), or 1 µM of Quinpirole (Tocris Biosciences) before a further round of imaging. For TTX (Sigma-Aldrich) measurements, organoids were incubated in 1 µM TTX for 30 min following baseline measurements before another 10 min of imaging.

Data analysis was performed using FIJI (National Institutes of Health, http://fiji.sc (accessed on 13 November 2023)) [85]. Regions of interest were manually selected using ROI manager, and mean fluorescence was calculated for each timeframe. Change in fluorescence was calculated as follows: ΔF/F = (F − F0)/F0, in which F0 was the mean fluorescence value recorded at t = 0, and a cell was considered electrically active if the fluorescence intensity change was greater than 0.2 ΔF/F. Cells were considered electrically active if they had at least one observable calcium spike during the 10-min period. For Figure 1E, Ca^2+^ transient frequency (number of calcium transients per minute) and amplitudes (ΔF/F) for individual cells were compared to their baseline frequency and amplitude before dopamine or ascorbic acid vehicle exposure. If there was a 20% increase in Ca^2+^ transient frequency or amplitude following compound exposure, cells were considered to have increased relative to the baseline, and if there was a 20% decrease, cells were considered to have decreased relative to the baseline. For all measurements *n* = 3 biological replicates of 3–5 dissociated and plated organoids were measured. For each biological replicate, the mean values for increase, decrease, or no change were calculated based on all cells that display calcium transients. Error bars for Figure 1E refer to the 95% confidence interval for the mean values for each individual biological replicate. Error bars for Appendix A refer to the 95% confidence intervals for the mean frequency and amplitude values across all cells within an individual biological replicate that display at least one calcium transient during the imaging period.

For cAMP imaging, the cADDis green down kit (Montana Biosciences) was used, and cells were prepared according to the manufacturer’s protocol. D83 organoids were dissociated using Accutase and plated on reduced growth factor Matrigel coated 24-well plates. Cells were cultured for 1 week in maturation media with 48-h media changes before dosing experiments were conducted. Frames were taken every 8 s for 20 min using a 10X objective. Dopamine was added at the 5-min mark with the preceding 5 min considered the baseline measurement. For all ROS inhibitor and dopamine receptor antagonist measurements, plated cells were treated with either 100 µM of acetylcysteine (Selleckchem) or 100 µM of D1 antagonist SCH-23390 (Tocris Biosciences) and Sulpiride (Tocris Biosciences) for 30 min prior to imaging.

Data analysis was performed using FIJI [85]. Regions of interest were manually selected using ROI manager, and mean fluorescence was calculated for each timeframe. Change in fluorescence was calculated as follows: ΔF/F = (F − F0)/F0, in which F0 was the mean fluorescence value recorded at t = 0. ΔF/F measurements were plotted over time, revealing an s-curve when the change in fluorescence decreased. The amplitude change was considered the linear region of the s-curve; the response time was calculated as the time after dopamine addition, before the initial decrease in fluorescence intensity—defined as the top of the linear section of the s-curve. For each biological replicate, a single ROI of interest encompassing the entire field of view was selected, as during this assay there were no individual cAMP transients. The fluorescence values were averaged within FIJI [85]. For all measurements n = 3 biological replicates of 3–5 dissociated and plated organoids were measured.

### 4.6. Dosing Experiments

To perform acute and chronic dopamine dosing for live cell imaging, plated D83 organoids were exposed to either 1 µM or 1 mM dopamine diluted in 100 µM of ascorbic acid or a 100 µM ascorbic acid vehicle control solution, which was added to the media directly. Each single dose involved addition of compound to the media for 1 h and then all the media was removed and replaced with fresh maturation media. Acute treatments constituted a single dose. Chronic treatment involved cells exposed to a single dose each day for 1 week. Acute samples and controls were time-matched to end at the same time as chronic samples, with prior dosing constituting daily single-dose treatments of ascorbic acid vehicle during the week leading up to imaging or RNA harvest. All samples were analyzed or harvested right at the end of the final single dose of the week.

For bulk sequencing and immunohistochemistry, D83 organoids were treated with either a dose of dopamine diluted in 100 µM of ascorbic acid, cocaine (NIDA drug supply program) diluted in 100 µM of ascorbic acid, or 100 µM ascorbic acid as vehicle control for 1 h before all the media was removed and replaced with fresh maturation media. Final concentrations of dopamine and cocaine were 1 µM while the concentration of ascorbic acid was kept constant. For chronic samples, organoids were dosed daily for 1 week before a final 1-h dose was administered. Acute samples and controls were dosed with ascorbic acid during the week prior before an acute dose of either dopamine or ascorbic acid was added. For all organoids treated with the ROS inhibitor, 100 µM of acetylcysteine was added to the media prior to dosing with dopamine or cocaine. For the final dose after 1 h of incubation, organoids were immediately washed once with cold PBS and fixed for sectioning and immunostaining or washed twice with cold PBS before samples were flash-frozen in liquid nitrogen and shipped on dry ice for RNA extraction and sequencing.

### 4.7. ROS Measurement

Oxidative stress experiments were modified from a protocol previously described [29]. Organoids were incubated with 100 μM 2′,7′-dichlorofluorescein diacetate (DCFH-DA) (Sigma-Aldrich) for 1 h along with dopamine, cocaine, or ascorbic acid control. Organoids were washed twice in PBS before being dissolved in 1% Triton in PBS. Fluorescence measurements were taken at an excitation wavelength of 485 nm and an emission wavelength of 530 nm using a Tecan plate reader. Protein concentrations were obtained using the BCA assay (Thermo Fisher) according to the manufacturer’s instructions. ROS levels were calculated by dividing the fluorescence measurements by the total protein concentration. All ROS values were then normalized to the vehicle controls.

### 4.8. RNA Extraction and Sequencing Analysis

Total RNA was extracted on site by the company performing sequencing. Sequencing and preparation of Illumina libraries was performed and then sequenced using Illumina Hi-seq 2 × 150 bp (Azenta, South Plainfield, NJ, USA or LC Sciences, Houston, TX, USA) for 20–30 million reads per sample. Raw sequencing data are publicly available on the Gene Expression Omnibus (GEO Accession Number: GSE234769).

Raw FASTQ formatted sequence reads were imported into CLC Genomics Workbench (v.21.0.5 Qiagen, https://digitalinsights.qiagen.com/ (accessed on 2 February 2023)). Adaptor sequences and bases with low quality were trimmed and reads were mapped to the reference genome (GRCh38.102) using the RNA-seq analysis tool with the default parameters recommended for RNA-seq analysis. Principal component analysis and differential expression analysis were performed using ‘PCA for RNA-seq’ and ‘Differential Gene Expression for RNA-seq’ toolsets. Differential gene expression was performed by comparing organoids either acutely or chronically dosed with dopamine or cocaine against the ascorbic acid dosed controls. Genes were considered differentially expressed if they had a log2FC > |1| and a false discovery rate (FDR) < 0.05.

Transcriptome profiles of 12 pcw striatum were downloaded from the BrainSpan database (www.brainspan.org (accessed on 15 March 2023)). Normalized gene expression values were averaged across the three replicates. Log_2_FC was calculated compared to normalized gene expression values.

For gene set enrichment analysis, all genes with an FDR < 0.05 were uploaded into Ingenuity Pathway Analysis (IPA) software [86] (Qiagen, https://digitalinsights.qiagen.com/IPA (accessed on 17 March 2023). Top regulated canonical pathways were predicted using −log_10_(*p*-value) while upstream regulators and comparisons were ranked based on z-score [86]. All plots and heatmaps were generated in Microsoft Excel (https://www.microsoft.com/en-us/microsoft-365/p/excel/cfq7ttc0hr4r?activetab=pivot:overviewtab (accessed January 8 2023)) and figures were compiled in Adobe Illustrator. Overlaps between gene sets were performed using the Venn diagram tool from the Bioinformatics & Evolutionary Genomics Department at Ghent University (https://bioinformatics.psb.ugent.be/webtools/Venn/ (accessed on 15 March 2023)). 

### 4.9. Statistical Analysis

Statistics were performed for live cell imaging and immunostaining quantification using one-way ANOVA with Tukey’s post hoc test to test significance. All analyses were performed using Microsoft Excel. Significance was defined as *p* < 0.05. To reduce clutter in indicating statistically significant comparisons, lettering was used for plots where ANOVA was used. Conditions in the same graph that do not share the same letter are statistically significant from one another. Statistical analyses for RNA-seq were performed in CLC Genomics Workbench (v.21.0.5 Qiagen, https://digitalinsights.qiagen.com/). All sequencing data passed default quality filters for the CLC Genomic Workbench version 21.0.5 RNA-Seq pipeline analysis. Significance for Ingenuity Pathway (v. 22.0, QIAGEN Inc., https://digitalinsights.qiagen.com/IPA, accessed on 13 January 2023) and analyses were defined as q-value < 0.05 and z-score > |2|.

## 5. Conclusions

In this study, we demonstrate that ventral forebrain organoids exhibit Ca^2+^, cAMP, and transcriptional responses to dopamine and cocaine [28]. Our results reveal that dopamine and cocaine induce these responses through ROS-mediated mechanisms, and that these responses are blunted upon ROS inhibition. In addition, transcriptomic analyses implicate immune, inflammation, ECM, and HOX genes in these responses. Overall, this work further motivates the use and continued development of ventral forebrain organoids as a valuable human model for studying responses to external stimuli [19,87,88].

## Figures and Tables

**Figure 1 ijms-24-16474-f001:**
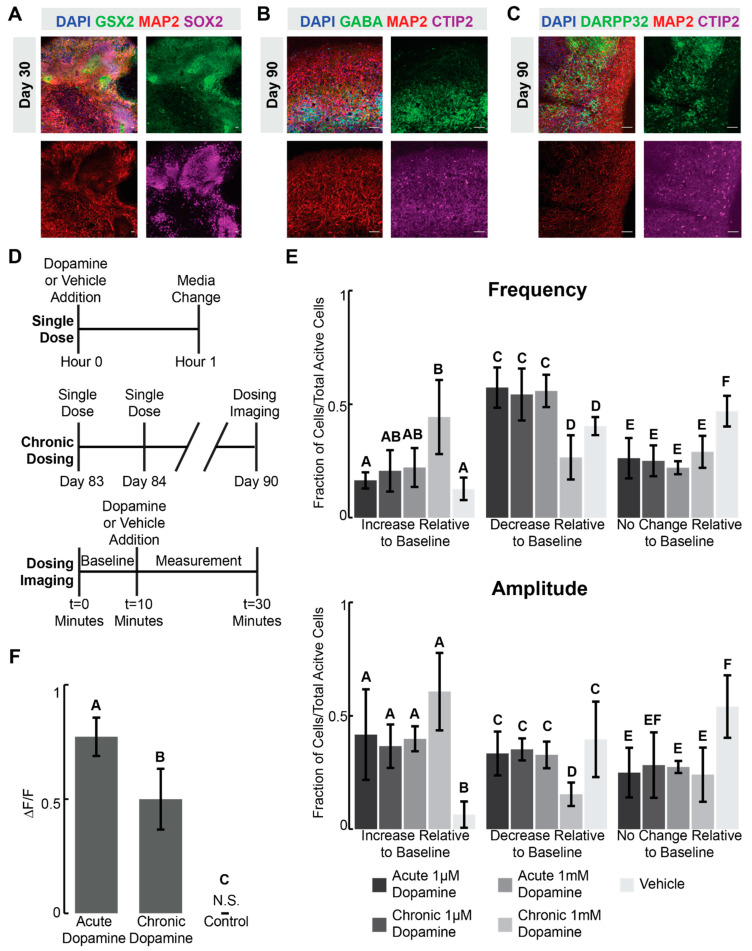
Ventral forebrain organoids exhibit Ca^2+^ and intracellular cAMP responses to dopamine (**A**–**C**) Immunostaining images of (**A**) day 30 or (**B**,**C**) day 90 (D90) Xiang–Tanaka organoids demonstrating expression of relevant ventral forebrain markers. (**A**–**C**) Scale bars = 100 μm. (**D**) Schematic illustrating what constitutes a single dose. Acute treatment is a single dose. Chronic treatment is a single dose applied daily for a week. (**E**) Ca^2+^ imaging frequency and amplitude measurements of individual neurons compared to their baseline controls following acute or chronic 1 µM or 1 mM dopamine exposure. An increase or decrease was defined as a cell having greater than 20% deviation from its baseline measurement following compound exposure. (**F**) Amplitude of intracellular cAMP change relative to baseline following acute or chronic 1 µM dopamine exposure. (**E**,**F**) Error bars represent 95% confidence intervals for n = 3 biological replicates, where each replicate is 3–5 organoids. (**E**) Each biological replicate represents the mean frequency and amplitude for all cells that displayed calcium transients during imaging experiments. (**F**) Biological replicate represents one measurement taken for all cells during the imaging experiment. Statistics: *p* < 0.05, one-way ANOVA with Tukey–Kramer post hoc analysis. Conditions sharing the same superscript letters are not statistically significant from each other. Therefore, if two conditions do not share the same lettering, they are statistically significant from each other. N.S. stands for non-significant.

**Figure 2 ijms-24-16474-f002:**
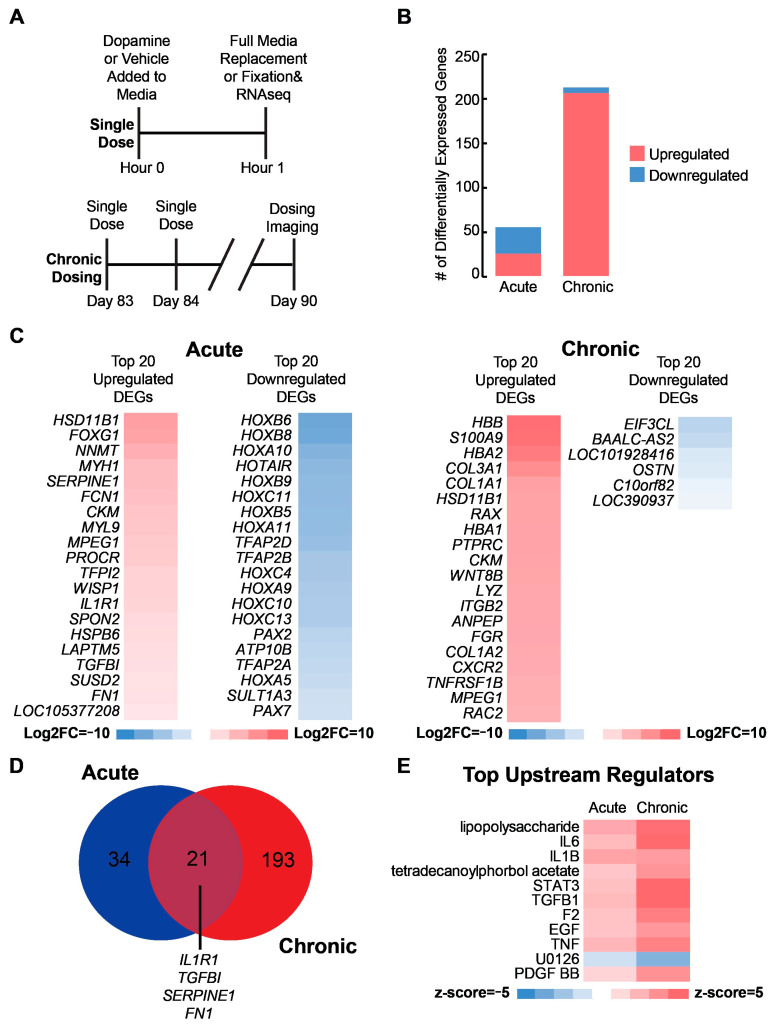
Transcriptional analysis reveals immune-related response to acute and chronic dopamine (**A**) Schematic illustrating what constitutes a single dose, and how daily single doses over a week comprise the chronic treatment condition used for the bulk sequencing experiment. A single dose refers to the addition of dopamine or vehicle control to the media before a full media change after 60 min. Dosing occurred once daily from D83 to D90 for chronic treatment. (**B**) Counts of differentially expressed genes (DEGs) following acute and chronic dopamine exposure. Differential gene expression (DGE) performed relative to vehicle controls. Genes are considered differentially expressed when they have log2FC > |1| and false discovery rate q < 0.05. (**C**) Heat map of the top 20 up and down regulated genes following acute or chronic dopamine exposure. (**D**) Venn diagram showing the overlap between acute and chronic DEGs. (**E**) Heat map of the top 10 regulated upstream regulators based on z-score using gene set enrichment analysis (GSEA). Regulator up- or downregulation is considered significant if z-score > |2|.

**Figure 3 ijms-24-16474-f003:**
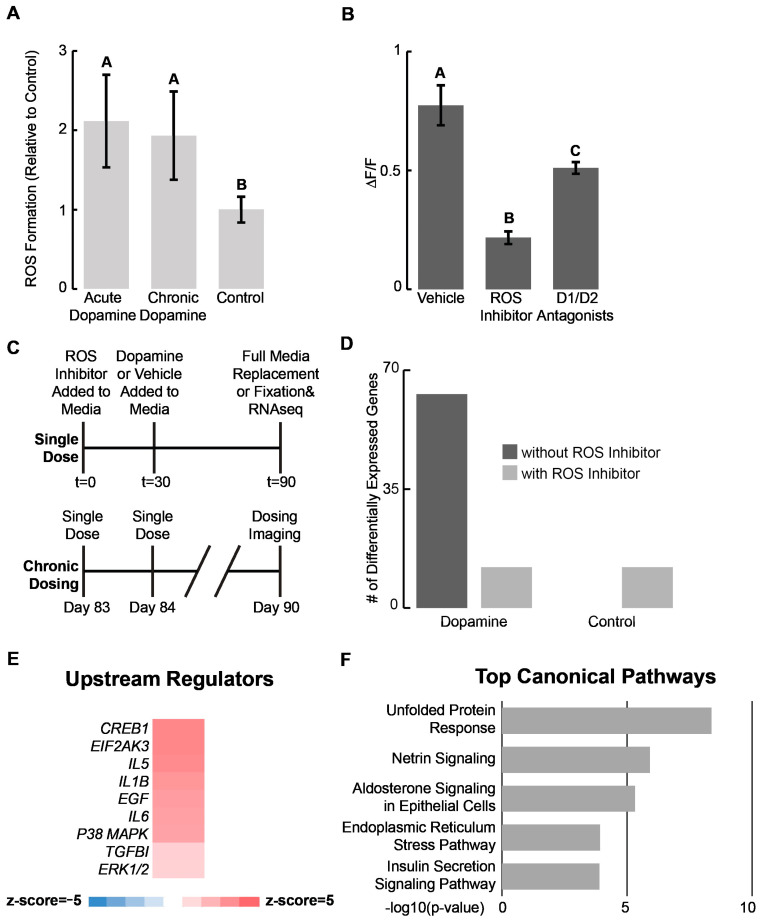
Organoid response to dopamine driven by reactive oxygen species (**A**) Reactive oxygen species in ventral forebrain organoids following acute and chronic dopamine exposure. Values normalized to vehicle controls. (**B**) Change in amplitude of intracellular cAMP signal relative to baseline following treatment with either vehicle, an ROS inhibitor, or D1/D2 dopamine receptor antagonists prior to acute dopamine exposure. (**A**,**B**) Error bars represent 95% confidence for n = 3 biological replicates. Statistics: *p* < 0.05, one-way ANOVA with Tukey–Kramer post hoc analysis. Conditions sharing the same superscript letters are not statistically significant from each other. Therefore, if two conditions do not share the same lettering, they are statistically significant from each other. (**C**) Schematic illustrating a single dose and chronic treatment used for bulk sequencing experiment. Time (t) in minutes. A single dose refers to the addition of dopamine or vehicle control to the media prior to a complete change of media after 60 min. Dosing occurred once daily from D83 to D90 for chronic treatment. (**D**) Number of DEGs following chronic dopamine exposure with or without an ROS inhibitor added 30 min prior to dopamine exposure. Differential gene expression performed relative to vehicle controls. Control with an ROS inhibitor added was compared against control without an ROS inhibitor. The dopamine dosed sample without an ROS inhibitor added was compared against the control without an ROS inhibitor added. The dopamine-dosed sample with an ROS inhibitor added was compared against the control sample with an ROS inhibitor added. (**E**) Heat map showing activation of select upstream regulator immune response genes calculated by z-score using GSEA. Regulator up- or downregulation is considered significant if z-score > |2|. (**F**) Top five regulated canonical pathways by −log10(*p*-value) identified using GSEA.

**Figure 4 ijms-24-16474-f004:**
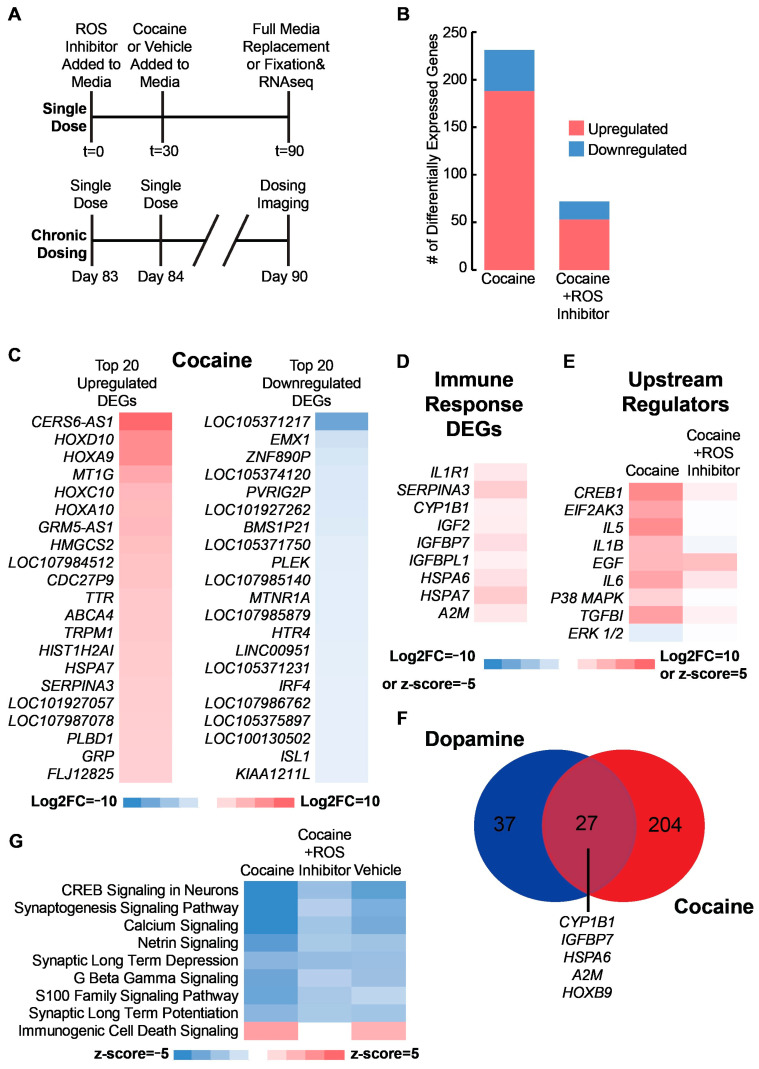
ROS also mediates the transcriptomic response to cocaine (**A**) Schematic illustrating what constitutes a single dose and chronic treatment used for the bulk sequencing experiment. Time (t) in minutes. A single dose refers to the addition of dopamine or vehicle control to the media before a full media change after 60 min. Dosing occurred once daily from D83 to D90 for chronic treatment. (**B**) Count of DEGs following chronic cocaine exposure with or without an ROS inhibitor. Differential gene expression performed relative to vehicle controls. The cocaine-dosed sample without an ROS inhibitor added was compared against the control without an ROS inhibitor added. The cocaine-dosed sample with an ROS inhibitor added was compared against the control sample with an ROS inhibitor added. (**C**) Heat map of top 20 up- and downregulated DEGs following cocaine exposure without an ROS inhibitor. (**D**) Heat map of immune response-related genes identified in Figure 3 following cocaine exposure without an ROS inhibitor. (**E**) Heat map of upstream regulators identified in Figure 3 following cocaine exposure with or without an ROS inhibitor. (**F**) Venn diagram showing the overlap between the DEGs following dopamine or cocaine exposure without an ROS inhibitor. (**G**) Heat map of the top regulated canonical pathways by z-score following dopamine or cocaine exposure with or without an ROS inhibitor.

## Data Availability

The data can be accessed at: GSE234769. Video files associated with Figure 1E,F are available at https://github.com/keung-lab/Rudibaugh-et-al-2023.git (13 November 2023).

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
