# Peer review of "Reactive Oxygen Species Mediate Transcriptional Responses to Dopamine and Cocaine in Human Cerebral Organoids"

_ijms, 2023, doi:10.3390/ijms242216474_

Round 1
Reviewer 1 Report
Comments and Suggestions for Authors
This paper was written by known experts in the field and further emphasizes the relevance of organoids in drug screening.
I have only minor edits suggestions:
1) I see no statistical significance in the text as well as figures- were the results not significant or was the sample number too low?
2) I would suggest quantifying the IF staining and adding an additional figure.
3) The discussion should also focus on inflammation since they have shown that the inflammatory pathway is upregulated in these conditions.
Reviewer 2 Report
Comments and Suggestions for Authors
This study constructed ventral forebrain organoids to model ventral forebrain neurodevelopment and investigated their responses to acute and chronic dopamine exposure using a combination of next generation sequencing along with Ca2+ and cAMP measurements. It is well executed that immune related transcriptional responses and identified novel response pathways activated following dopamine exposure. These findings demonstrated the utility of ventral forebrain organoids as an in vitro human platform for studying complex biological processes and their potential mechanisms.
All-in-all, I have just minor-to-medium comments.
Comments:
1. The abstract looks like it needs to be revised by reducing the background presentation, incorporating more detailed methods and results, and highlighting the focus of the study.
2. For calcium imaging, it is better for the authors to give a real-time video recording of calcium ion fluorescence to show its electrically active and observable calcium spike.
3. Please give the name of the software (FIJI) used for calcium imaging data analysis and the company that produced it, or its website.
For example: ImageJ software (National Institutes of Health, https://imagej.nih.gov/ij/)
4. In the calcium imaging experiments, D83 organoids were dissociated using Accutase and plated on reduced growth factor Matrigel (Corning)coated 24 well plates. Why do organoids need to separate into single cells here? After replantation, are cells in 2D or 3D condition? Please describe it in the methods.
